# Inferring the size distribution of volcanic ash from IASI measurements and optimal estimation

Luke M. Western<sup>1</sup>, Peter N. Francis<sup>2</sup>, I. Matthew Watson<sup>1</sup>, and Shona Mackie<sup>3</sup> <sup>1</sup>COMET, School of Earth Sciences, University of Bristol, Bristol, UK <sup>2</sup>Met Office, Exeter, UK <sup>3</sup>School of Earth Sciences, University of Bristol, Bristol, UK *Correspondence to:* Luke M. Western (luke.western@bristol.ac.uk)

**Abstract.** This study demonstrates a method of retrieving the mass column loading and cloud-top pressure of a volcanic ash cloud, together with the effective radius and spread of the ash particle size distribution, as well as the cloud top pressure of any underlying water cloud, using an optimal estimation technique applied to Infrared Atmospheric Sounding Interferometer data. Two shapes of particle size distribution are considered, a log-normal and a gamma distribution. Results show that it is viable

- 5 to retrieve a measure of the size distribution spread, namely the geometric standard deviation, when a log-normal distribution is assumed, whereas this is not the case for an assumed gamma distribution in terms of its effective variance. The volcanic conditions under which the method works well are discussed, as are its shortcomings. The method is applied to two volcanic eruptions: Eyjafjallajökull, Iceland using data from 6th May 2010 and Kasatochi, Alaska using data from 8th August 2008. The results show that the retrieved geometric standard deviation of these ash clouds is spatially variable, and is generally similar
- 10 to what is assumed in many passive infrared remote sensing techniques. An abrupt change in the retrieved geometric standard deviation has been observed for the Eyjafjallajökull eruption along the trajectory of the ash cloud, and possible explanations for this are discussed.

# 1 Introduction

Recent years have seen advancement in passive infrared remote sensing of volcanic ash. This was encouraged, at least in part,
by the widespread disruption caused during the 2010 eruption of Eyjafjallajökull, Iceland. Improvements in ash detection, (e.g. Clarisse et al., 2013; Mackie and Watson, 2014) and quantification of ash cloud properties (e.g. Francis et al., 2012; Pavolonis et al., 2013) has allowed for corroboration (Millington et al., 2012), insertion (Wilkins et al., 2014) and assimilation (Pelley et al., 2015) of remotely sensed data with atmospheric dispersion models to assist Volcanic Ash Advisory Centres (VAACs) in preparing their advisories.

Despite progress in the field of volcanic ash remote sensing, large uncertainties are still apparent in many retrieval schemes (Corradini et al., 2008; Francis et al., 2012). In particular, little attention has been paid to the uncertainties encountered due to uncertainty in particle size distribution, or the spread of the distribution when the shape of the particle size distribution is known (Western et al., 2015). It is unlikely that the shape of the particle size distribution is known prior to retrieval and conflicting data can emerge from in situ measurement from a single event (Turnbull et al., 2012). Issues surrounding airborne

5

10

in situ measurement are not unique to volcanic ash studies, see for example the discussion on measurement of meteorological ice particles in Baumgardner et al. (2012).

This paper presents a method using the hyperspectral satellite sensor Infrared Atmospheric Sounding Interferometer to infer an approximate size distribution of a volcanic ash cloud in terms of the moments of the size distribution and the limitations of such a method. Some preliminary conclusions on particle size distribution are presented and compared to current knowledge.

2 Infrared Atmospheric Sounding Interferometer

The Infrared Atmospheric Sounding Interferometer (IASI) is a nadir-looking Fourier transform spectrometer onboard the Metop satellites (Clerbaux et al., 2009). The satellites have a polar sun-synchronous orbit with twice daily earth coverage and a 50 km field of view at nadir with 4 footprints of 12 km in diameter over a swath width of  $\sim$ 2200 km. The main advantage of the IASI instrument is its unprecedented spectral coverage of 645 to 2760 cm<sup>-1</sup>, with spectral sampling of 0.25 cm<sup>-1</sup> and

apodized spectral resolution of 0.5  $\rm cm^{-1}$ .

The IASI data used in the current study are acquired from the UK Centre for Environmental Data Archival (CEDA) after processing at level 1c.

# 3 Method

The method employed here is, in essence, an amalgamation of the work by Francis et al. (2012) and Clarisse et al. (2010) with a measure of the spread of a particle size distribution added to the state vector used for the retrieval. Both these works, and the current method, follow the work of Rodgers (2000) on inverse methods and optimal estimation. The approach taken, as applied to volcanic ash retrieval, is briefly described in the following.

#### 3.1 Optimal estimation

The basis of optimal estimation is the minimisation of the cost function defined by:

$$-2\ln P(\mathbf{x} \mid \mathbf{y}) = [\mathbf{y} - \mathbf{F}(\mathbf{x})]^T \mathbf{S}_{\epsilon}^{-1} [\mathbf{y} - \mathbf{F}(\mathbf{x})] + [\mathbf{x} - \mathbf{x}_a]^T \mathbf{S}_a^{-1} [\mathbf{x} - \mathbf{x}_a],$$
(1)

where y is the observation, in this case the measured spectral radiances,  $\mathbf{F}(\mathbf{x})$  is the simulated observation, in this case the radiance calculated by some forward model,  $\mathbf{S}_{\epsilon}$  contains the covariances between the observation and forward model, x is the state vector, containing the parameters to be retrieved,  $\mathbf{x}_a$  is the vector containing the a priori estimates of the state vector and

$S_a$  is the covariance between the state vector and the a priori estimates, i.e. the assumed uncertainty in knowledge of the state vector.

The minimisation is reached by an iterative method where the state vector is updated using Levenberg-Marquadt iterations of the form:

$$\mathbf{x}_{n+1} = \mathbf{x}_n + [(1+\gamma)\mathbf{S}_a^{-1} + \mathbf{K}_n^T \mathbf{S}_\epsilon^{-1} \mathbf{K}_n]^{-1} \{\mathbf{K}_n^T \mathbf{S}_\epsilon^{-1} [\mathbf{y} - \mathbf{F}(\mathbf{x}_n)] - \mathbf{S}_a^{-1} [\mathbf{x}_n - \mathbf{x}_a]\},\tag{2}$$

where  $\gamma$  is the Marquadt step parameter and K denotes the Jacobian matrix. In this case the scaling matrix has been chosen as  $\mathbf{S}_a^{-1}$ . The most likely solution of the state vector is reached when the gradient of the cost function equals zero, or when some convergence criterion is reached. The gradient of the cost function is given as:

$$\nabla_{\mathbf{x}} \{-2\ln P(\mathbf{x} \mid \mathbf{y})\} = -\mathbf{K}^T \mathbf{S}_{\epsilon}^{-1} [\mathbf{y} - \mathbf{F}(\mathbf{x})] + \mathbf{S}_a^{-1} [\mathbf{x} - \mathbf{x}_a],$$
(3)

where  $\nabla_{\mathbf{x}}$  denotes the Fréchet derivative. 5

#### 3.2 Application to retrieval of volcanic ash

The state vector is chosen to encompass the retrievable parameters of an ash cloud, with elements consisting of the ash cloud top pressure  $p_{ash}$ , the ash mass column loading L, the effective radius of the particle size distribution  $r_e$  and the spread of the particle size distribution S, namely the geometric standard deviation  $\sigma_g$  or effective variance b of a log-normal or

- gamma distribution respectively. This is a simplification of the cloud microphysical properties where assumptions on the ash 10 composition, sphericity and conformation to an analytical particle size distribution are made. A log-normal distribution and gamma distribution have both been suggested as representative particle size distributions for volcanic ash clouds (Wen and Rose, 1994; Prata and Grant, 2001; Clarisse et al., 2010; Francis et al., 2012) and the retrieval method is run separately for both size distributions. For a review of in situ measurements of airborne size distributions see Western et al. (2015). Ash clouds often
- have underlying low-level stratus cloud present (see e.g. Prata and Prata, 2012, Fig. 7), which affects the upwelling radiation 15 incident on the ash cloud. For this reason, a further element is included in the state vector,  $p_{\rm cld}$ , which is the cloud top pressure of an underlying water cloud and where a value of  $p_{\rm cld}$  equal to the surface pressure corresponds to the case of an ash cloud having no underlying water cloud. This underlying water cloud is assumed to be a blackbody. This is deemed a reasonable approximation and simplifies the problem of scattering between multiple cloud layers (Watts et al., 2011).

20

The resultant representative state vector is therefore defined as:

$$\mathbf{x} = egin{bmatrix} p_{\mathrm{ash}} \ L \ r_e \ S \ p_{\mathrm{cld}} \end{bmatrix}.$$

A log-normal distribution can be characterised by the probability density function:

$$n(r) = \frac{N_0}{\sqrt{2\pi}r\ln\sigma_g} \exp\left(-\frac{\left(\ln r - \ln r_g\right)^2}{2\ln^2\sigma_g}\right),\tag{4}$$

where  $N_0$  is the total number of particles, r is the particle radius and  $r_q$  is the geometric mean radius of the distribution.

The effective radius of a distribution of particles is defined as:

$$r_e = \frac{\int_0^\infty r^3 n(r) dr}{\int_0^\infty r^3 n(r) dr},\tag{5}$$

which for a log-normal distribution simplifies to:

$$r_e = r_g \exp\left(\frac{5}{2}\ln^2\sigma_g\right). \tag{6}$$

Conserving notation, and using *b* to represent the effective variance (Hansen and Travis, 1974), the probability density function for a gamma distribution is given as:

$$n(r) = N_0 \frac{(r_e b)^{\frac{2b-1}{b}}}{\Gamma(\frac{1-2b}{b})} r^{\frac{1-3b}{b}} \exp\left(-\frac{r}{r_e b}\right).$$

$$\tag{7}$$

A simple forward model is applied to simulate the IASI radiances (Prata, 1989), where the vector  $\mathbf{F}(\mathbf{x})$  is made up of modelled radiances  $I_i^{\text{mod}}$  for channel *i*:

$$I_i^{\text{mod}}(\mathbf{x}) = (1 - \varepsilon_i) I_i^{\text{up}}(p_{\text{cld}}) + \varepsilon_i I_i^{\text{oc}}(p_{\text{ash}}),$$
 (8)

where  $I_i^{up}$  is the modelled top of atmosphere upwelling radiance with no ash cloud present and  $I_i^{oc}$  is the modelled top of atmosphere overcast radiances due to a blackbody at pressure  $p_{ash}$ .  $\varepsilon_i$  corresponds to the modelled spectral emissivity if an ash cloud and is given by:

$$\varepsilon_i = 1 - \exp\left(-\frac{k_{\text{ext}}(r_e, S)L}{\cos\theta}\right).$$
(9)

$k_{\text{ext}}$  is the scaled mass extinction coefficient (Chou et al., 1999) of a distribution of particles of size  $r_e$  and spread  $\sigma_g$  or b calculated using a Mie scattering routine (Bohren and Huffman, 2008) and  $\theta$  is the satellite zenith angle.

Upwelling and overcast radiances are forward modelled using the fast radiative transfer code RTTOV (Hocking et al., 2013) at the IASI pixel location. Surface and atmospheric data are taken from the European Centre for Medium-Range Weather Forecasts (ECMWF) ERA-Interim Global Reanalysis data set (Dee et al., 2011) on 60 model levels. The ECMWF data are interpolated linearly in both space and time to IASI pixel locations and acquisition times.

It would be computationally impractical and would not greatly increase information content to use all IASI channels (Collard, 2007). For this reason a subset of 100 channels are selected in the range 700-1000 and 1100-1200  $\text{cm}^{-1}$  every 4  $\text{cm}^{-1}$ , where the ozone absorption band between 1000-1100  $\text{cm}^{-1}$  has been excluded. These channels have been chosen as they are in an atmospheric window sensitive to volcanic ash, incorporating the carbon dioxide absorption band to assist in the constraint of

25 pressure at ash cloud top.

20

The a priori values used in retrieval are:

$$\mathbf{x}_{a} = \begin{bmatrix} 500 \text{ hPa} \\ 5 \text{ g m}^{-2} \\ 3 \mu \text{m} \\ 2.0, 0.15 \\ p_{\text{surf}} \end{bmatrix},$$

where the fourth row is the a priori geometric standard deviation or effective variance assumed for the particle size distribution respectively and  $p_{surf}$  denotes the surface pressure, i.e. no underlying water cloud is present in the a priori state.

The standard deviation values assumed for the covariance matrix  $S_a$  are 500 hPa for ash cloud top pressure, 10 g m<sup>-2</sup> for mass column loading, 7 µm for effective radius, 1.0 and 0.1 for geometric standard deviation and effective variance respectively and 200 hPa for the underlying water cloud top pressure. The square of these values form the diagonal of the covariance matrix  $S_a$  and should be representative of the large uncertainty in the selection of a priori values.

The covariances between the forward modelled and IASI observations have been found using a method similar to that in
Carboni et al. (2012). A local covariance matrix has been formed by finding the covariance between observed and modelled radiances for clear-sky scenes in an area where the ash cloud is present but during a time when no eruptive event has occurred. Clear-sky pixels are identified using the method described by Mackie et al. (2010) for days in January, April, July and October during a year in which no proximal eruption has taken place. This should account for uncertainties from observation, the forward model, atmospheric and surface profile data and any atmospheric constituents otherwise unaccounted for (e.g. thin overlying cloud cover).

The elements of the Jacobian matrix  $\mathbf{K} = \nabla_{\mathbf{x}} \mathbf{F}(\mathbf{x})$  can be calculated as follows:

$$\frac{\partial I_i^{\text{mod}}(\mathbf{x})}{\partial p_{\text{ash}}} = \left\{ 1 - \exp\left(-\frac{k_{\text{ext}}(r_e, S)L}{\cos\theta}\right) \right\} \frac{\partial I_i^{\text{oc}}(p_{\text{ash}})}{\partial p_{\text{ash}}},\tag{10}$$

$$\frac{\partial I_i^{\text{mod}}(\mathbf{x})}{\partial L} = \frac{k_{\text{ext}}(r_e, S)}{\cos\theta} (I_i^{\text{oc}}(p_{\text{ash}}) - I_i^{\text{up}}(p_{\text{cld}})) \exp\left(-\frac{k_{\text{ext}}(r_e, S)L}{\cos\theta}\right),\tag{11}$$

$$\frac{\partial I_i^{\text{mod}}(\mathbf{x})}{\partial r_e}\Big|_S = \frac{L}{\cos\theta} (I_i^{\text{oc}}(p_{\text{ash}}) - I_i^{\text{up}}(p_{\text{cld}})) \exp\left(-\frac{k_{\text{ext}}(r_e, S)L}{\cos\theta}\right) \left.\frac{\partial k_{\text{ext}}(r_e, S)}{\partial r_e}\right|_S,\tag{12}$$

$$\frac{\partial I_i^{\text{mod}}(\mathbf{x})}{\partial S}\Big|_{r_e} = \frac{L}{\cos\theta} (I_i^{\text{oc}}(p_{\text{ash}}) - I_i^{\text{up}}(p_{\text{cld}})) \exp\left(-\frac{k_{\text{ext}}(r_e, S)L}{\cos\theta}\right) \left.\frac{\partial k_{\text{ext}}(r_e, S)}{\partial S}\right|_{r_e},\tag{13}$$

$$\frac{\partial I_i^{\text{mod}}(\mathbf{x})}{\partial p_{\text{cld}}} = \exp\left(-\frac{k_{\text{ext}}(r_e, S)L}{\cos\theta}\right) \frac{\partial I_i^{\text{up}}(p_{\text{cld}})}{\partial p_{\text{cld}}}.$$
 (14)

(15)

#### 3.3 Information content of method

The information content of the retrieval in relation to the elements of the state vector can be represented by the averaging kernel, given by:

$$\mathbf{A} = (\mathbf{K}^T \mathbf{S}_{\epsilon}^{-1} \mathbf{K} + \mathbf{S}_a^{-1})^{-1} \mathbf{K}^T \mathbf{S}_{\epsilon}^{-1} \mathbf{K}.$$

This averaging kernel represents the sensitivity of the retrieval of the state vector elements to the physical state of what is to be retrieved.

The trace of the averaging kernel can be interpreted as the degrees of freedom for signal of the retrieval  $(DOF = \sum a_{ii})$ . This quantity is representative of the number of elements of the state vector that can be retrieved given a physical state, albeit a continuous rather than discrete quantity. The degrees of freedom for a range of ash cloud properties for a log-normal

- distribution and no underlying water cloud have been calculated and can be seen in Fig. 1 and for a gamma distribution in Fig. 2. The degrees of freedom for an ash cloud with an underlying water cloud can be seen in Fig. 3 and Fig. 4, where the parameters describing the ash cloud are compared to a low-level underlying water cloud of varying altitude. For each grid, two elements are varied across a range of likely encountered state vector values while the remaining state vector elements are kept constant as their a priori values  $x_a$ . Here an atmospheric profile was taken from 6th May 2010 (59°N 12°W) and  $S_{\epsilon}$  was
- produced locally as described in the previous section. The refractive index used has been measured from ash collected during the 2010 Eyjafjallajökull eruption (Dan Peters, *personal communication*).

Figures 1 - 4 suggest that, in general, the retrieval of the spread of a log-normal particle size distribution has greater information content than a gamma particle size distribution in relation to the physical state of the atmosphere and ash cloud due to the greater number of physical states with high degrees of freedom. It is reasonable to conclude that one would be unable to detect

- changes in the effective variance of a gamma distribution with much conviction in most volcanic remote sensing situations. However, a change in the geometric standard deviation of a log-normal particle size distribution appears to be detectable in many conditions that would be of practical use in volcanic remote sensing applications. The limitation in the detection of a change in size of particle size distribution with a large effective radius is known for broadband passive infrared retrievals (Wen and Rose, 1994; Pavolonis et al., 2013) and this limitation is also apparent in hyperspectral retrievals due to the inherent single
- scattering properties of volcanic ash. Other limitations appear to be when the ash cloud is of low altitude ( $p_{ash} \gtrsim 700 \text{ hPa}$ ), when a size distribution spread is large, and when the column loading has very low or high values.

Figure 5 shows the fourth element of the diagonal of the averaging kernel, i.e. the element that describes the degree of freedom for signal of the retrieval of spread for (a) a log-normal and (b) a gamma size distribution, for different values of effective radius. The ash cloud is assumed to have its ash cloud top at the tropopause, no underlying water cloud and a mass

loading of 5 g m<sup>-2</sup>. These plots reinforce the difficulty in measuring a change in spread for large effective radii and a large value for spread.

It should be noted that the averaging kernel is strongly influenced by the choice of  $S_a$ . It follows that a change in choice for the elements of  $S_a$  could change the interpreted degrees of freedom for signal. Airborne volcanic ash has not been measured as having a gamma size distribution in situ, whereas a log-normal distribution has been measured (Farlow et al., 1981; Schumann

et al., 2011; Johnson et al., 2012). Other studies may have used a gamma size distribution due to it being measured for other volcanic species, such as sulphuric acid (e.g. Brogniez et al., 1997). For this reason only a log-normal particle size distribution is used hereafter in the retrieval scheme to approximate the spread of the ash particle size distribution.

# 3.4 Optical properties of ash

- The mass extinction coefficient is known to vary with the refractive index and spread of a distribution of particles (Jennings et al., 1979; Bohren and Huffman, 2008). This means that mass extinction is both wavelength and size distribution dependent, which leads to a unique spectral signal when a broad range of wavelengths is used. The dependence can be quite clearly seen for a log-normal distribution in Fig. 6, where geometric standard deviations between  $\sigma_g = 1.5 3.0$  are shown at wavelengths of 13.4, 12.0, 10.8 and 8.7 µm using refractive indices for ash from the 2010 Eyjafjallajökull eruption (Dan Peters, *personal*
- *communication*). This serves as the physical basis as to how a unique size distribution can be retrieved over a broad spectral range. This uniqueness is much less pronounced for a gamma distribution (Fig. 7), where a change in effective variance *b* is much less distinguishable at most effective radii and relevant wavelengths, and hence fewer degrees of freedom. For Figs. 6 and 7 the spreads of size distributions are intended to be representative of those likely to occur in nature for volcanic aerosols.

# 4 Case studies

# 15 4.1 2010 Eyjafjallajökull eruption

The April - May 2010 Eyjafjallajökull eruption in Iceland was well observed and has been extensively studied from a variety of perspectives (e.g. Schumann et al., 2011; Gudmundsson et al., 2012; Prata and Prata, 2012; Spinetti et al., 2013). Since the eruption a refractive index data set has been measured for ash collected in Iceland on 17th April 2010 (Dan Peters, *personal communication*) and the bulk density of the very fine ash deposited in the period 4-8th May has been determined to be

$2738 \text{ kg m}^{-3}$  (Bonadonna et al., 2011). These data would most likely be unavailable during any real-time monitoring of an eruption but is convenient for post-eruptive analysis. The particle size distribution has been shown to be log-normal through in situ aircraft measurement (Schumann et al., 2011; Johnson et al., 2012), although discrepancies in the spread of the distribution exist (Turnbull et al., 2012).

Figure 8 shows the brightness temperature difference of an IASI image, with an overpass at 21:50 UTC on 6th May 2010. The brightness temperature difference is the difference between brightness temperatures at 10.8 and 12.0  $\mu$ m (926 and 833.5 cm<sup>-1</sup> respectively),  $BTD = BT_{926} - BT_{833.5}$ . Volcanic ash generally gives a negative BTD while water or ice cloud generally has a positive BTD (Prata, 1989).

# 4.2 2008 Kasatochi eruption

Kasatochi is a volcano located in the central Aleutian Islands of Alaska. Between 7 - 8th August 2008 it underwent a violent 30 eruption where the majority of the ash was deposited over sea. There is little available compositional information on the distal

5

ash from the eruption, however it can be inferred from proximal juvenile material that the eruption was andesitic in composition (Waythomas et al., 2010) and an andesitic refractive index (Pollack et al., 1973) has been used for the retrieval. Although the assumption of refractive index is necessary, it is expected that the detection (Mackie et al., 2014) and retrieval (Clarisse et al., 2010; Francis et al., 2012) is sensitive to the refractive index chosen. The ash bulk density is assumed to be similar to other andesitic eruptions and a value of  $2300 \text{ kg m}^{-3}$  has been used (Bonadonna and Phillips, 2003). The size distribution of the volcanic ash is an unknown and a log-normal particle size distribution has been chosen as the most suitable.

The eruption plume contained a large SO<sub>2</sub> loading (Prata et al., 2010) and can be seen by the large  $\nu_1$  absorption band present around 1152 cm<sup>-1</sup> in Fig. 9. The forward model used in the retrieval, Eq. (8), does not account for the presence of SO<sub>2</sub> in the ash cloud. It was therefore decided to only use channels in the range 700-1000 cm<sup>-1</sup> for the retrieval from this eruption.

10 The brightness temperature difference of an IASI image from 8th August 2008, with an overpass at 21:25 UTC, can be seen in Fig. 10.

# 4.3 Detection of ash

For this work, pixels containing volcanic ash have been identified using a probabilistic detection scheme (Mackie and Watson, 2014), whereby the probability with which a pixel is associated with each of three states, clear, cloud and ash, is calculated.
The states are assumed to be mutually exclusive and so the probabilities for any one pixel sum to unity. Mixed cloud-ash pixels are likely to correspond to elevated probabilities for both cloud and ash, while clear pixels and 'pure' ash and cloud pixels are likely to correspond to probabilities approaching either zero or unity for each state. For the purposes of this work, pixels with a calculated probability for ash exceeding 0.2 were deemed to contain ash, and all other pixels were deemed ash free.

#### 5 Results and Discussion

# 20 5.1 Retrieved properties

The properties of ash clouds, from the 2010 eruption of Eyjafjallajökull on 6th May and 2008 eruption of Kasatochi on 8th August have been retrieved using the method described in section 3 and the IASI sensor. The retrieved state vector values of the ash cloud can be seen in Figs. 11 and 12 for Eyjafjallajökull on 6th May and Kasatochi on 8th August repectively. The underlying cloud top pressure is not shown as it is not a parameter that describes the ash cloud itself, rather a nuisance

25 parameter of necessity to better describe the system. The effects of including an underlying water cloud in the retrieval is an interesting problem in itself as changes in the upwelling radiation incident on the ash cloud will affect retrieved parameters. The inclusion of a non-clear-sky atmosphere to model upwelling radiation during retrieval may be important for improvements to retrievals using both broadband and hyperspectral sensors.

# 5.2 2010 Eyjafjallajökull eruption retrieval

A histogram showing the retrieved geometric standard deviation values from Fig. 11(a) can be seen in Fig. 13(a). Two modes are apparent around  $\sigma_q = 2.0$  and 2.8, the implications of which are discussed in the section 5.4.

# 5.3 2008 Kasatochi eruption retrieval

- Unlike the 2010 eruption of Eyjafjallajökull, no refractive index data are available for the 2008 Kasatochi eruption and the composition has been inferred from juvenile material deposited mainly through pyroclastic density currents, which may result an incorrect assumption for the refractive index. Similarly, no in situ data are available on the shape of the particle size distribution of the ash. The size distribution conforming to a log-normal shape is merely an assumption to which the entire ash cloud may not conform.
- The geometric standard deviation values for Kasatochi from Fig. 12(a) are shown in Fig. 13(b), which has a single mode present around  $\sigma_g = 2.1$ .

### 5.4 Discussion

Particle sedimentation from an ash cloud is somewhat complex, where particle settling and vertical transfer is more sophisticated than the Lagrangian dynamics of individual particle settling (Carazzo and Jellinek, 2013). Diffuse convection can lead

to such phenomena as particle layering and fingering during sedimentation (Green, 1987; Osborn et al., 1995; Carazzo and Jellinek, 2013; Costa et al., 2013). As a result, it seems reasonable that the size distribution of an ash cloud, in terms of spread, does not remain temporally or spatially constant in a heterogenous atmosphere.

The retrieved geometric standard deviation for Eyjafjallajökull from 6th May 2010 has a mode around 2.0, in line with current assumed geometric standard values (e.g. Clarisse et al., 2010), and also consistent with values derived from in situ sampling during the eruption, where geometric standard deviation values in the range 1.8 - 2.5 were measured (Turnbull

- sampling during the eruption, where geometric standard deviation values in the range 1.8 2.5 were measured (Turnbull et al., 2012). A larger mode around 2.8 is also present, a value greater than those observed by in situ measurement during the eruption. The geometric standard deviation appears to get smaller as the ash cloud progresses. In situ measurement during the 2010 Eyjafjallajökull eruption was carried out in areas of relatively low mass concentration. These ash clouds were generally more aged than that shown in Fig. 11 (Schumann et al., 2011; Johnson et al., 2012). If the geometric standard deviation of the
- particle size distribution is indeed changing along the trajectory, it is therefore of little surprise that in situ observations differ from those retrieved in Fig. 13(a).

There is a narrowing in the spread of the size distribution of the Eyjafjallajökull ash cloud at around 14°W 55°N and westward from here. This appears to match a change in the altitude of the ash cloud. This could be for a variety of reasons and some of the possibilities are suggested.

<sup>(1)</sup> As previously mentioned, atmospheric sorting can lead to layering of particles. Here the lower altitude ash cloud farther along the trajectory could be a lower altitude layer containing finer particles after such sorting effects have taken place.