# Peer review of "Inferring the size distribution of volcanic ash from IASI measurements and optimal estimation"

_Atmospheric Measurement Techniques, 2016_

## Referee Comment (RC1) · Anonymous Referee #1 · 29 May 2016

This paper deals with the quantification of airborne volcanic ash from infrared satellite observations. The authors attempt to simultaneously retrieve five parameters, but the emphasize of the paper is clearly on the retrieval of two size parameters: the effective radius and the spread of the psd. Retrieval of the latter is the innovative aspect of the paper, as most other satellite retrieval schemes retrieve only the effective radius. The first part of the paper analyzes the information content of IASI spectra using forward simulations, especially again with respect to the size parameters. The second part then illustrates the retrieval on two case studies. While the topic is worthwhile and the paper is well written, the text and analysis is very short and lacks depth in several places. In my opinion the main result sections (3.3 and 5) need to be reworked/expanded and

reviewed again before the paper can be considered for publication.

Section 3.3 on the information content relies on the averaging kernel being a good measure of information content. However, as the authors point out, the averaging kernel depends on the a priori covariance matrix, which was chosen here to be (very) large; e.g. $x_a = 500$ hPa and $S_a = 500^2$ hPa$^2$ for the aerosol cloud top pressure . The DOF, in the Rogers formalism expresses the gain of information, and if one starts out with almost no information (large Sa), the gain will always be large. Looking at the limiting case $S_a \rightarrow \infty$ and thus $S_a^{-1} \rightarrow 0$, we have that $A \rightarrow I$. Therefore, getting large values (here  5) of the DOF is not so difficult if one starts out with a large Sa; but this does not necessarily mean that the information is contained within the spectrum. All of this should be discussed in more detail. I would also urge the authors to find a better way of presenting/analyzing the information content. Perhaps either by looking at the relative increase of the DOF, what I believe was suggested by the other referee in the technical report, or to quantify the information content in another way, e.g. looking at the retrieval error covariance matrix, which is easier to interpret than the averaging kernel, and still provides information for large S$_a$.

Figures 1 to 5 are almost not discussed, although there is potentially a lot to talk about. Without proper discussion, there is really no point in having them. It is not easy to present results on a 2 x 5D space, and it would perhaps be easier not to discuss the underlying cloud case or the gamma distribution (which does not seem to bring anything to the table anyway). Discussing the triplet "effective radius, spread, mass" and how information on these three parameters can or cannot be independently retrieved would be much more interesting. Looking at the on and off-diagonal elements of the retrieval error covariance matrix and its corresponding correlation matrix would be useful.

The presented case studies are not very convincing, and again not sufficiently discussed or explored. Please choose a different colormap for figures 11 and 12, the different shades of green are really hard to distinguish. The colormap used in figures

1-4 would do. Some of the retrieved values seem very large. If the results are what they seem, I really doubt they make sense. Very large values seem to indicate that the retrievals were not sufficiently constrained. But again, it is practically impossible to read the data from the colormap and apart from the spread, all the other retrieval results are also not discussed at all in the text. It would also help if some of the retrieval results (e.g. altitude) are compared with external information (e.g. CALIOP or other satellite retrievals). Another way to deepen the analysis and impact of the paper would be to repeat the retrieval for a fixed lognormal spread and subsequently compare the retrieved masses and heights (with and without retrieving the spread).
* * *

---

## Referee Comment (RC2) · Anonymous Referee #3 · 8 Jul 2016

The paper "Inferring the size distribution of volcanic ash from IASI measurements and optimal estimation" by Luke M. Western et al. presents a new optimal estimation technique to retrieve different parameters of the ash size distribution, loading and altitude, as well as the altitude of eventually present underlying clouds. Sensitivity analyses and information content estimations are discussed in the first part of the paper, while two test cases are discussed in the second part. The case studies are focussed on the retrieval of the effective radius and the spread of an assumed log-normal size distribution. The retrieval of the spread of the size distribution is probably the more innovative aspect of this work (while the simultaneous retrieval of underlying clouds and their altitudes is also very interesting but not tackled in depth in the present study). The paper is

well written. In my opinion, the paper might be ideally worthwhile but a number of major issues should be solved before it can be published. I strongly encourage the Authors to tackle these issues because the topic of this paper is very interesting. Please find in the following some Major Comments (MC) and Specific and Technical Comments (SC). Page and Lines of the manuscript are indicated in the individual comments (Px, Ly).

Major Comments:

1) As already suggested by another Referee, the manuscript is very concise. This is normally a good thing but in this case several aspects are not sufficiently discussed and need to be expanded. Two very clear cases where the Authors must enhance their discussion is Sect. 3.4 (e.g., why not showing the Dan Peters refractive index? See also MC3) and Sect. 5.4 (e.g., why not associating to Fig.13 the distribution of the effective radius, and then discuss the size distribution evolution using arguments for both the effective radius and distribution spread variability?) A few other examples are in the Specific Comments section.

2) Again, as suggested by another Referee, the exclusive analysis of the averaging kernels is not sufficient at all to quantify the information content of the observations/method. The complete information content quantification is usually performed using both the averaging kernels and the retrieval errors. In the case of this methodology, I suspect that, due to the very large values of the a priori covariance matrix, the fairly high DOF could be accompanied by relatively large retrieval errors. Please calculate and discuss the retrieval errors.

3) The optical properties of ash and their impact on the observed brightness temperatures depend critically on the refractive index assumed for the ash, and then from its chemical/mineralogical composition. While this is mentioned in the text, a more detailed discussion must be given. Section 3.4 must appear before. Sensitivity analyses on the relative importance of size distribution parameters and the chemical composi-
tion of ash should be performed and discussed. How can the reader know if a lack of knowledge in the chemical composition/refractive index of the ash would introduce enough uncertainties in the retrieval of size distribution parameters to render these retrieved quantities unusable?

4) While the possible simultaneous presence of SO2 in the plume is mentioned (and caution is taken to avoid the SO2-sensitive spectral region in the Kasatochi case study), nothing is said about the possible presence of sulphate aerosols and their impact on the ash retrieval. Sulphate aerosols are known to have an extinction (in particular due to absorption, for real world situations) signature in the range 800-1200 cm-1 [Sellitto and Legras, 2016]. In the IASI spectrum for Kasatochi shown in Fig. 9, apart from the very clear SO2 signature at 1100-1200 cm-1, a quite typical sulphate aerosols signature can be quite probably seen (increasing extinction from 800 to 1200 cm-1). Limiting your input spectra at 700-1000 cm-1 for this case (as said in Sect. 4.2) would probably limit sulphate aerosols impact but this must be discussed. The possible impact of sulphate aerosols should be also discussed elsewhere in the paper when interfering species are mentioned.

Specific and technical comments:

1) In the Introduction and maybe in the abstract it should be mentioned why the retrieval of the spread of the ash particle size distribution is important (e.g., it has an impact on the ash dispersion? It has an impact on the local and regional radiative forcing? It has an impact on the cloud microphysics?)

2) In the Introduction and maybe in the abstract it should be mentioned that the simultaneous retrieval of underlying clouds and their altitude is useful for local/regional radiative forcing estimations (i.e., the aerosol radiative forcing depends on the reflectivity of the underlying surface).

3) The IASI acronym should be introduced also in the abstract

**AMTD**
4) Abstract, P1, L8: "Alaska"  $\rightarrow$  "Alaska (USA)"

5) P1, L8-10: the sentence "The results...techniques" is contradictory. You affirm that you have found that the spread of the size distribution is spatially variable and that it is similar to what assumed in other retrieval techniques (where it is assumed as spatially constant). Please rephrase.

6) P1, L12: "...possible explanations for this are discussed.", maybe you can briefly mention the proposed explanations?

7) Introduction, P1, L14: "...of volcanic ash."  $\rightarrow$  "...of volcanic ash and its spatial distribution and dispersion"

8) P1, L15-16: "Improvements in ash detection, (e.g., Clarisse et al., 2013)...", I'm not sure that this is the most pertinent citation here. This paper does not actually talks about "improvements in ash detection" in particular.

9) P1, L20: "large uncertainties", can you discuss a little bit these uncertainty estimations (how they are estimated and their value).

10) P1, L23: you cite Western et al., 2015 but I would discuss a little bit what is useful in that paper for the present manuscript. Can you mention what is said about the "uncertainties encountered due to the uncertainty in particle size distribution..."?

11) P2, L3: here and elsewhere, please change "hyperspectral" to "high spectral resolution", as the definition of "hyper-", "multi-" etc is quite subjective.

12) P2, L3: introduce the IASI acronym here and not at L6.

13) Why not integrating Sect. 2 into the "Method" Section?

14) P2, L8: I would say more "MetOp" than "Metop". Please mention MetOp A, B and C.

15) P2, L11: "0.5 cm-1"  $\rightarrow$  "0.50 cm-1"
16) P2, L13: please define "level 1c"

17) Section 3, P2, L15-17: please consider to invert the two sentences, i.e. first telling that you use an optimal estimation method and then telling that you merge Francis and Clarisse methods.

18) P2, L23: "some forward model"  $\rightarrow$  "some forward radiative transfer model"

19) P2, L25; "the assumed uncertainty"  $\rightarrow$  "the assumed a priori uncertainty"

20) P3, L1: "...the Jacobian matrix.", please put a sentence to define the Jacobian matrix

21) P3, L1: "...the scaling matrix...", please define the scaling matrix

22) P3, L10-11: "This is a simplification...are made": as composition and sphericity/asphericity are not retrieved and are assumed in your method, how can you be sure that the assumption you make are OK? The composition of ash can be greatly variable and the assumption of sphericity is highly questionable. How can you know that uncertainties linked with these assumptions do not destroy the sensitivity of the observations to the spread of the size distribution, e.g.? This must be thoroughly discussed and maybe further sensitivity analyses to these parameters (or published uncertainties estimations) must be performed. Section 3.4 might be moved here and used as a starting point for this discussion.

23) P4, L8: the reference "Prata, 1989" is for the simplified model used here? If so, please move this reference to "A simple forward radiative transfer model (Prata, 1989) is applied to..."

24) P4, L16: do you use a Mie scattering routine from Bohren and Huffman (there are Fortran codes in this book) or rather you use another software? Please specify.

25) P4, L22-23: "For this reason...has been excluded", of course it should be mentioned the more systematic micro-windows selection procedure of Echle et al., 2000.
26) P4, L23: "...the ozone absorption...", there are other interfering parameters (e.g. sulphate aerosols, see MC4, water vapour, etc) that must be discussed here.

27) P5, L2-4: the reasons for choosing these a priori values, especially for ash, L and re must be discussed with relevant literature.

28) P5, L10-11: " A local...has occurred", this is not clear at all to me. How can an ash cloud be present with no eruption? Please rephrase.

29) P5, L14-15: "thin overlying cloud cover", do you mean "thin cirrus clouds"?

30) P5, Equations 10-14: is it really necessary to show these equations?

31) P6, L1: "Information content of method"  $\rightarrow$  "Information content"

32) Section 3.3: Please calculate and discuss also the retrieval errors (MC2).

33) P6, L8: "...that can be retrieved..."  $\rightarrow$  "...that are independent..."

34) P6, L9-11: "The degrees of freedom...Fig. 2", the sentence is a little bit clumsy, please rephrase

35) P6, L19-20: "It is reasonable...situations", this is a poorly justified statement: please discuss more

36) P6, L23: "change in size of particle size" -> "change in width of particle size"?

37) P6, L 27-31: and what if there are underlying clouds?

38) P7, L2: "sulphuric acid", you might mean "sulphate aerosols"

39) P7, L3: please suppress "to approximate...distribution."

40) P7, Section 3.4 should appear well before (e.g., in Section 3.2)

41) P7, L5: I would rather say that the extinction coefficient can be calculated with Mie theory and, following this theory, it varies with the refractive index and the size distribution.

AMTD
42) P7, L 6: "This means...dependent" -> "This means that mass extinction is both wavelength, size distribution and chemical composition dependent" (the dependency on chemical composition must be mentioned here)

43) P7 or elsewhere: if the Dan Peters' refractive index is unpublished, please provide a plot and mention a chemical/mineralogical composition which can be associated with this refractive index

44) P7, 13: "...likely to occur in nature for volcanic aerosols.", references should be provided to justify this statement.

45) P7, L26: please define before the BTD you are using (maybe giving it a unique acronym, e.g. BTD(ash) or something).

46) P8, L2: "andesitic refractive index (Pollack et al., 1973)": which differences of this refractive index with respect to the Peters one? Please discuss it more in details.

47) P8, L7-9: please discuss the possible impact of sulphate aerosols extinction [Sellitto and Legras, 2016] (MC4).

48) P9, Section 5.4: the Authors discuss the variability of the spread of the size distribution but the variability of the effective radius should be discussed as well.

49) Figure 1-4: it is clear that the DOF can vary between 0 and 5 but why not changing the colour scale to better exploit its dynamics (most of the DOF values are between 1.5 and 4.5)?

50) Figure 3: I do not understand the meaning of the points at high ash pressure and low underlying cloud pressure. For example, for an ash pressure of 1000 hPa and a cloud pressure of 200 hPa, it is actually an overlying cloud"? Please explain.

51) Figures 11-12: please change the colour scale, as this green gradient scale is quite illegible.

52) Figure 13: please associate histograms for the effective radius and discuss them,
together with the spread.

References:

Echle et at al., 2000, "Optimized spectral microwindows for data analysis of the Michelson Interferometer for Passive Atmospheric Sounding on the Environmental Satellite.", Appl Opt 39 (2000) 5531-5540

Sellitto and Legras, 2016, "Sensitivity of thermal infrared nadir instruments to the chemical and microphysical properties of UTLS secondary sulfate aerosols", Atmos. Meas. Tech., 9, 115-132, doi:10.5194/amt-9-115-2016.